# Inactivation of *HAP4* Accelerates *RTG*-Dependent Osmoadaptation in *Saccharomyces cerevisiae*

**DOI:** 10.3390/ijms24065320

**Published:** 2023-03-10

**Authors:** Maria Antonietta Di Noia, Pasquale Scarcia, Gennaro Agrimi, Ohiemi Benjamin Ocheja, Ehtisham Wahid, Isabella Pisano, Eleonora Paradies, Luigi Palmieri, Cataldo Guaragnella, Nicoletta Guaragnella

**Affiliations:** 1Department of Biosciences, Biotechnologies and Environment, University of Bari “Aldo Moro”, 70125 Bari, Italy; 2Department of Electrical and Information Engineering, Politecnico di Bari, 70125 Bari, Italy; 3Institute of Biomembranes, Bioenergetics and Molecular Biotechnologies, National Research Council, 70126 Bari, Italy

**Keywords:** *RTG* signaling, Hap complex, osmoadaptation, mitochondria, TCA cycle, respiratory competence, stress response, inter-organelle communication, metabolism

## Abstract

Mitochondrial *RTG* (an acronym for ReTroGrade) signaling plays a cytoprotective role under various intracellular or environmental stresses. We have previously shown its contribution to osmoadaptation and capacity to sustain mitochondrial respiration in yeast. Here, we studied the interplay between *RTG2*, the main positive regulator of the *RTG* pathway, and *HAP4*, encoding the catalytic subunit of the Hap2-5 complex required for the expression of many mitochondrial proteins that function in the tricarboxylic acid (TCA) cycle and electron transport, upon osmotic stress. Cell growth features, mitochondrial respiratory competence, retrograde signaling activation, and TCA cycle gene expression were comparatively evaluated in wild type and mutant cells in the presence and in the absence of salt stress. We showed that the inactivation of *HAP4* improved the kinetics of osmoadaptation by eliciting both the activation of retrograde signaling and the upregulation of three TCA cycle genes: citrate synthase 1 (*CIT1*), aconitase 1 (*ACO1*), and isocitrate dehydrogenase 1 (*IDH1*). Interestingly, their increased expression was mostly dependent on *RTG2*. Impaired respiratory competence in the *HAP4* mutant does not affect its faster adaptive response to stress. These findings indicate that the involvement of the *RTG* pathway in osmostress is fostered in a cellular context of constitutively reduced respiratory capacity. Moreover, it is evident that the *RTG* pathway mediates peroxisomes–mitochondria communication by modulating the metabolic function of mitochondria in osmoadaptation.

## 1. Introduction

Mitochondria play multifaceted roles in cell stress response, acting not only as energetic hubs for eukaryotic aerobic metabolism but also as signals transducers to maintain cell homeostasis and cell fate [1,2]. In this regard, the emerging concept of mitochondrial cross-talk with other organelles, such as peroxisomes, endoplasmic reticulum (ER), and lysosomes, in the regulation of cell fate and function represents an additional cue [3,4,5]. Thus, understanding the mechanisms underlying the integration of mitochondria within the cellular context of stress signaling pathways and inter-organelles communication is relevant and needs further investigation. Yeast is a suitable eukaryotic model to study how mitochondria can have an impact on cellular response under physio-pathological conditions, thanks to the well-known evolutionary conservation of genes and pathways.

Mitochondrial retrograde signaling is a communication pathway from mitochondria to nucleus that is well-studied in cells with mitochondrial dysfunction, such as rho0 petite [6,7]. *RTG* (an acronym for ReTroGrade) genes are the major positive regulators of mitochondrial retrograde signaling, with *RTG2* acting as the main upstream regulator and *RTG1* and *RTG3* forming a heterodimer, binding the promoters of *RTG*-target genes. The upregulation of the peroxisomal isoform of citrate synthase (*CIT2*) is considered prototypical of *RTG* pathway activation. In fact, its main role is to replenish the tricarboxylic acid (TCA) cycle, compromised of rho0 cells, to sustain the transcriptional reprogramming required for re-establishing metabolic homeostasis [8,9]. 

Mitochondrial retrograde pathway has also been shown to be involved in different kinds of stress [5,10,11,12,13]. In particular, the determinant role of mitochondria in osmotic stress has been highlighted, although the molecular details are far from being elucidated [14,15,16]. Recently, we reported that the *RTG* pathway acts downstream of *HOG1*, the master regulator of osmostress response, and sustains mitochondrial respiratory capacity upon salt stress [13,17,18]. 

In this work, we studied the impact of *HAP4* inactivation on osmoadaptation and its interplay with *RTG2. HAP4* encodes the regulatory subunit of the Hap2-5 complex, which is considered a master regulator of mitochondrial function by coordinating the expression of nuclear and mitochondrial genes and encoding subunits of the respiratory chain complexes, components of the mitochondrial translation apparatus, mitochondrial import, and division that define the metabolic state of the cell [19,20,21]. The Hap complex and the *RTG* pathway can activate the same target genes in response to diauxic shift or compromised mitochondrial function, and a first indication of their possible interaction has been reported in [8]. Our data show that *HAP4* disruption improved the kinetics of osmoadaptation, and this was due to the specific upregulation of TCA cycle genes (citrate synthase 1, *CIT1*; aconitase 1, *ACO1;* and isocitrate dehydrogenase 1, *IDH1*), whose transcription appeared also controlled by the *RTG* pathway. Obtained results suggest that *HAP4* can modulate *RTG* pathway function and that the partial restore of TCA cycle activity, independent of mitochondrial respiratory capacity, is important for faster osmoadaptation.

## 2. Results

### 2.1. Deletion of HAP4 Accelerates Osmoadaptation in a Manner Dependent on RTG2 

It is well-established that mitochondria are important determinants of efficient stress adaptation. To gain insight into the role of mitochondrial function in cellular osmostress response, wild type (WT) and mutant yeast cells lacking HAP4 (hap4Δ), RTG2 (rtg2Δ), and both genes (hap4Δrtg2Δ) were analyzed by spotting assay for cell growth with glucose as the sole carbon source in the presence and in the absence of NaCl as stressor. It has been previously demonstrated that the concentration of 0.8 M NaCl is associated with an intermediate salt stress phenotype [13]; thus, we chose this concentration for all the experiments.

As shown in Figure 1, while WT cells were sensitive to the stress after two days of growth, HAP4 deletion appeared to improve cell fitness. Interestingly, deletion of RTG2 in the hap4Δ background abolished the increased stress resistance. As expected, when only RTG2 was absent, an increased sensitivity to NaCl stress was observed. In the absence of the stressor, all cells grew similarly under these experimental conditions. The improved growth phenotype of hap4Δ was observed exclusively after two days of growth, suggesting a faster kinetics of adaptation. To better define growth and survival on-plate at different conditions and times, we applied a quantitative imaging-based protocol measuring cell density within single spots of defined size from spotting assay image (Figure 1). Growth of 100% corresponded to the cell density measured in the absence of the stress. After two days, the growth percentage of WT and hap4Δ cells on NaCl plate was about 45% and 60%, respectively, compared to about 10% in the double mutant and single RTG2 mutant. The growth difference between WT and hap4Δ was abolished after three days, while the relative growth of hap4∆rtg2∆ and rtg2Δ increased about three times (30%). These data demonstrate that the abrogation of HAP4 function accelerates the cellular response to osmostress.

To further explore this phenotype, we analyzed cell growth in both micro- and macro- (shake flask) cultures up to 24 h (Figure 2). After 8 h, relative growth in micro-cultures was about 27 and 47% comparing WT and hap4Δ, respectively (Figure 2A). These significant differences were maintained also at the next time points. hap4∆rtg2∆ and rtg2Δ showed reduced growth compared to WT and hap4Δ, especially after 16 h. The doubling time was calculated for each cell type and experimental condition, showing a significant decrease for hap4Δ compared to WT upon NaCl stress, 3.0 versus 3.9 h (Table 1). No differences in doubling time were observed between hap4∆rtg2∆ and rtg2Δ in NaCl, 5.95 versus 5.9 h. Results from shake flask cultures were comparable to micro-cultures, demonstrating a higher relative growth of hap4Δ cells compared to WT at each time analyzed, and growth decrease when RTG2 was deleted with HAP4 (Figure 2B). 

### 2.2. Mitochondrial Respiratory Competence Is Not Required for Faster Adaptation 

We have previously shown that RTG2 sustains mitochondrial respiration in osmoadaptation [13]. To gain insight into the role of mitochondrial function in stress response, we assessed mitochondrial respiratory competence in WT and mutant cells. For this purpose, the cells were grown in YPD with or without NaCl and spotted after 5 h on YP plates with glucose or glycerol as fermentative and not-fermentative carbon sources, respectively. As shown in Figure 3, WT and rtg2Δ were able to grow on glycerol, indicating respiratory capacity after the stress. However, cell growth was severely compromised in hap4Δ and hap4Δrtg2Δ. Similar results were obtained when the cells were grown in the absence of stress and then spotted on YPD/YPGly plates. These data indicate that respiratory competence was maintained in WT and rtg2Δ but constitutively impaired in hap4Δ and hap4Δrtg2Δ. When the cells were grown on glucose as the sole carbon source, no differences in viability among strains could be seen. After 24 h, reduced respiratory competence could be observed in glycerol growing rtg2Δ compared to WT cells (data not shown), in accordance with [13]. 

These data confirm that HAP4 is the main regulator of mitochondrial respiratory competence independent of the stress conditions or RTG2, which becomes relevant to sustain mitochondrial respiration only in a late phase of the stress. Moreover, it is evident that mitochondrial respiratory competence is not essential for faster osmoadaptation, and its suppression is believed to represent an advantage. 

### 2.3. RTG Pathway Is Activated in Cells Lacking HAP4 

We have previously shown that RTG pathway is activated by NaCl [13]; thus, to gain more insight into the molecular mechanisms supporting the growth advantage of hap4Δ upon salt stress, the expression of CIT2 was analyzed in all cell types with or without stress (Figure 4). About a four-fold CIT2 increase was observed in the absence of HAP4 upon stress. This upregulation was comparable to WT cells and, as expected, absent in RTG2-lacking mutants. CIT2 expression was further downregulated in hap4Δrtg2Δ cells. 

These data demonstrate that RTG pathway is upregulated upon NaCl treatment in the absence of HAP4 and confirm its major role in osmotic stress response. 

### 2.4. HAP4 and RTG2 Modulate Mitochondrial TCA Cycle Genes Expression upon Salt Stress 

It has been previously reported that mitochondrial TCA cycle enzymes increased upon osmoshock and that the RTG pathway controls the expression of selected TCA cycle genes in cells with compromised respiratory function [8,14]. Thus, to gain insight into the mechanisms of faster osmoadaptation, the expression of these genes (CIT1, ACO1, and IDH1) was analyzed after 5 h in WT and in mutant cells lacking HAP4 or RTG2 in the absence or presence of stress (Figure 5). It is noteworthy that CIT1 expression was significantly upregulated by NaCl in hap4Δ cells and was conversely downregulated in WT and rtg2Δ in the same conditions (Figure 5A). The expression of ACO1 and IDH1 reached very high levels in hap4Δ cells treated with NaCl compared with relative controls and WT cells, in which a downregulation due to the stress was observed in both genes. (Figure 5B,C). In cells lacking RTG2, the expression of these two genes was increased in the presence of NaCl, although to a lesser extent than that observed in hap4Δ (Figure 5B,C). 

Overall, these data show the interplay between HAP4 and RTG2 in the modulation of mitochondrial TCA cycle gene expression upon salt stress. 

## 3. Discussion

The relationship between mitochondrial function and cellular stress response by using yeast as a model highlighting the relevance of metabolic signatures in the dynamics of adaptive response has been explored in this study.

We showed that *HAP4*, the master regulator of mitochondrial function coordinating nuclear and mitochondrial gene expression, and *RTG2*, the upstream positive regulator of the *RTG* pathway, reciprocally interact to control mitochondrial function upon salt stress. 

Specifically, the inactivation of *HAP4* confers an advantage in the kinetics of osmoadaptation, as judged by cell growth analysis and parameters both on solid and in liquid micro- and macro-cultures (Figure 1 and Figure 2). The data obtained clearly showed that faster stress response depends on the *RTG* pathway, whose role in metabolic reprogramming and osmoadaptation is reinforced [13,23,24]. However, neither the mitochondrial respiratory capacity nor activation of the *RTG* pathway itself seem to be determinant in the dynamics of osmoadaptation (Figure 3 and Figure 4). In this scenario, peroxisomes–mitochondria cross-talk and mitochondrial metabolic reprogramming appear to be crucial (Figure 5 and Figure 6). In fact, a concomitant increase in the expression of *CIT2* and of the first three genes of TCA cycle (*CIT1*, *ACO1,* and *IDH1*) was measured in stressed cells lacking *HAP4* with respect to WT (Figure 6). These genes support anabolic reactions through the production of TCA cycle intermediates, such as citrate and oxoglutarate, precursors for NADPH and amino acids, respectively. A stress-specific and strictly *RTG*-dependent upregulation could be observed for *CIT1*, suggesting a possible role played by citrate in faster osmoadaptation. In this regard, an increase in citrate levels coming from both the peroxisomal and the mitochondrial sources might be hypothesized. Mitochondrial citrate could be exported by the mitochondrial carrier Yhm2p, whose physiological role is to increase the NADPH-reducing power in the cytosol and to act as a key component of the citrate-oxoglutarate NADPH redox shuttle between mitochondria and cytosol [25]. Cytosolic NADPH could exert an antioxidant function against oxidative damage caused by NaCl and improve adaptation [14]. Alternatively, citrate in the cytoplasm could enter mitochondria in exchange with oxoglutarate to be used for glutamate biosynthesis [26,27]. According to this hypothesis, *hap4Δ* mutant, differently from *rtg2Δ* and *hap4Δrtg2Δ*, was able to grow in the absence of glutamate on minimal medium in the presence of NaCl (data not shown).

On the other hand, the upregulation of the aconitase-encoding gene *ACO1* in both *HAP4* mutant and *RTG2* mutant, although at a lesser extent, beyond its metabolic significance might highlight the importance of mitochondrial DNA in osmoadaptation. This hypothesis is supported by previous data showing the direct effect of *ACO1* expression on mitochondrial DNA maintenance [28]. Overall, these data agree with a role for peroxisomes in supporting mitochondrial function and ensuring stress survival [3]. 

The results also demonstrate that an impairment of respiratory function coupled to *RTG*-mediated mitochondrial metabolic reprogramming can accelerate osmoadaptation. In this context, *HAP4* appears to be a negative regulator in the kinetics of stress response, which agrees with recent studies showing that reduction in *HAP4* protein levels may be advantageous to achieving mitochondrial homeostasis by coordinating gene expression between the mitochondrial and nuclear genomes in respiratory-deficient cells [29].

Future work will be directed to the elucidation of upstream signals and mechanisms of mitochondria-mediated osmoadaptation by studying the impact of inter-organellar cross-talk and mitochondrial dysfunctions on cellular stress response. 

*Hog1*, the main regulator of osmostress, induces a rapid and transient cytoplasmic response that acts on glycerol accumulation and transport through the aquaglyceroporin Fps1p. *RTG* signaling is activated in the second line of stress defense, as determined by *CIT2* upregulation, to sustain peroxisome–mitochondria cross-talk and mitochondrial metabolic function. *HAP4* might act directly or indirectly as a negative regulator of *RTG*-dependent expression of TCA cycle genes *CIT1*, *ACO1*, and *IDH1.*

## 4. Materials and Methods

### 4.1. Yeast Strains and Growth Conditions 

The *S. cerevisiae* strains used in this study were W303-1B (WT) cells (MATα *ade*2 *leu*2 *his*3 *trp1 ura*3) and derivatives *hap4∆* (*hap4*∆::*KanMX4*), *rtg2∆* (*rtg2*∆::*LEU2*), and *hap4∆rtg2∆* (*rtg2*∆::*LEU2 hap4*∆::*KanMX4*) obtained as described in [10]. Cells were grown at 30 °C in YPD (1% yeast extract, 2% bactopeptone (GIBCO, Life Technologies, Waltham, MA, USA), and 2% glucose (Sigma-Aldrich, St. Louis, MO, USA) with 2% agar (Invitrogen, Life Technologies Waltham, MA, USA) for solid medium in the absence or in the presence of 0.8 M sodium chloride (NaCl). Cell growth was monitored qualitatively on YPD agar plates and quantitatively by measuring optical density (600 nm) on liquid YPD medium cultures grown either in micro-well plates or in flasks. Mitochondrial respiratory competence was assessed by spotting equal amounts of serially diluted cells treated or not with NaCl on YPD and YPGlycerol solid [30]. 

### 4.2. Micro- and Batch-Culture Growth Assays 

For micro- and batch-culture growth assays, fresh yeast cells cultured for about 16 h at 30 °C were diluted in triplicate in multi-well plates or flasks to the same initial OD600. Optical density was then constantly monitored using a high-precision TECAN microplate reader equipped with a shaker and a temperature control unit or by using a Thermo Spectronic Genesis 20 spectrophotometer at selected times. Micro-culture growth curves were analyzed in Microsoft Excel and doubling time was calculated as reported in Toussaint et al. [31]. Relative growth was calculated in both micro- and batch-cultures as the percentage of the optical density values under stress conditions (with NaCl) compared to the control (without NaCl) at different times. At least three independent cultures were analyzed for each condition in each independent experiment. 

### 4.3. Spotting Assay 

Fresh overnight yeast cultures (30 °C, YPD medium) were adjusted to the same optical density (OD600 = 1), and 5 microliters of each serial dilution was spotted on YPD agar medium with or without NaCl. Plates were incubated at 30 °C for 2–5 days, and images were acquired by means of a ChemiDoc Touch Imaging System and analyzed using Image Lab software. 

The images obtained by spotting assay were quantitatively analyzed by using the protocol described in Petropavlovskiy et al. [22]. Relative cell growth was calculated comparing the cell density numerical values obtained from each selected spot dilution in control and treated cells. Growth of 100% corresponded to the cell density of each spot measured in the absence of the stress. 

To better define the growth and survival on-plate at different conditions and times, we applied a quantitative imaging-based protocol based on the measurement of cell density within single spots of defined size on an image of spotting assay (Figure 1B). Growth of 100% corresponded to the cell density measured in the absence of the stress. 

### 4.4. Quantitative PCR (qPCR) 

The mRNA levels of peroxisomal citrate synthase-encoding gene (*CIT2*), mitochondrial citrate synthase-encoding gene (*CIT1*), aconitase (*ACO1*), and isocitrate dehydrogenase (*IDH1*) were determined in continuously growing cells after 5 h of NaCl exposure and in the absence of stress. Then, 5 × 10^7^ cells were collected and centrifuged at 3000× *g*. Cell pellets were stored at −80 °C before total RNA extraction with a Presto Mini RNA Yeast Kit (Geneaid, New Taipei City, Taiwan). We immediately performed 0.5 µg RNA (OD260/OD280 ≥ 2.0) reverse transcribed using a QuantiTect Reverse Transcription Kit (Qiagen, Hilden, Germany), and cDNA was directly used for quantitative PCR (qPCR) analysis or stored at −20 °C. The actin 1 (*ACT1*) mRNA was amplified in parallel and used as a housekeeping reference gene. The QuantStudio 3 Real-Time PCR System (Applied Biosystems, Thermo Fisher Scientific, Waltham, MA, USA) was used for Quantitative PCR using the primer pairs (Table 2) based on the cDNA sequences of the investigated genes and designed with Primer Express 3.0 (Applied Biosystems, Thermo Fisher Scientific, Waltham, MA, USA). The primers were purchased from Invitrogen (Thermo Fisher Scientific, Waltham, MA, USA). Twenty microliters of reaction volume contained 20 ng of reverse-transcribed first-strand cDNA, 10 µL of SYBR Select Master Mix (Applied Biosystems, Thermo Fisher Scientific, Waltham, MA, USA), and 300 nM of each primer. The specificity of the PCR amplification was checked with the heat dissociation protocol after the final cycle of PCR. 

To correct for differences in the amount of starting first-strand cDNAs, the ACT1 mRNA was amplified in parallel as a reference gene. The relative quantification of the investigated genes was performed according to the comparative method (2^−ΔΔCt^) [27]. 2^−ΔΔCt^ = 2^−(ΔCt sample−ΔCt calibrator)^, where ΔCt sample is Ct sample − Ct reference gene and Ct is the threshold cycle. For the calibrator, ΔΔCt = 0 and 2^−ΔΔCt^ = 1. The value of 2^−ΔΔCt^ indicates the fold change in gene expression relative to the calibrator.

### 4.5. Statistical Analysis 

All the experiments were repeated at least three times, and the results are reported as means with standard deviation. For the determination of significant differences between samples, all results were analyzed using Student’s *t*-test run on Microsoft Excel software, with significant differences indicated as *p* values in the range between 0.05 and 0.0001. 

## Figures and Tables

**Figure 1 ijms-24-05320-f001:**
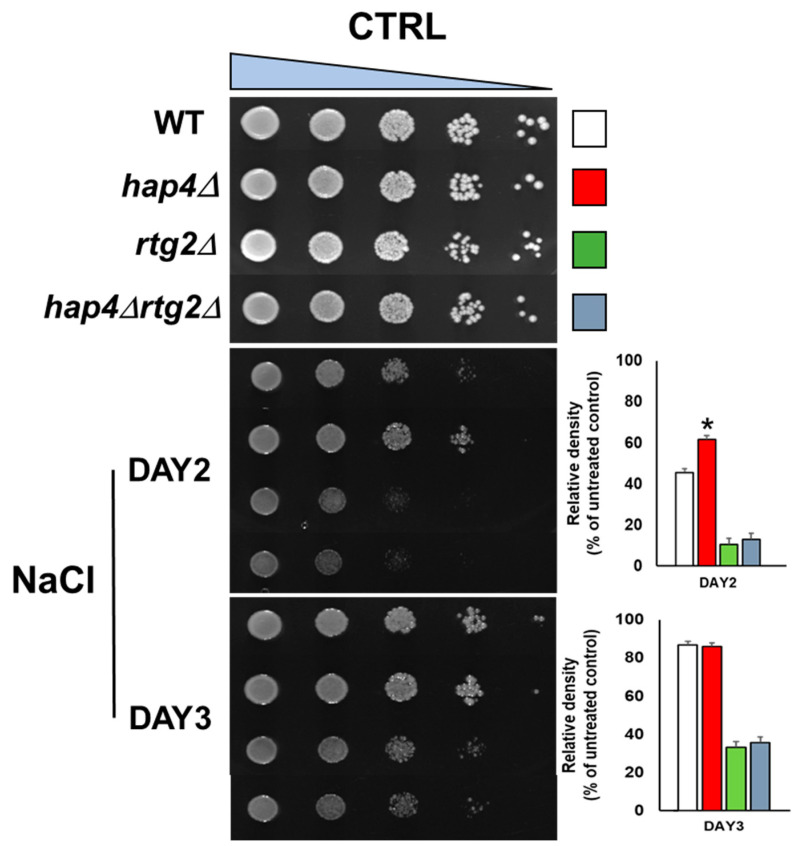
Sensitivity of wild type and mutant cells to sodium chloride. Wild type (WT) and mutant cells, lacking HAP4 (hap4∆), RTG2 (rtg2∆), or HAP4 and RTG2 (hap4∆ rtg2∆), were grown overnight in YPD medium and diluted to 1 OD600, and ten-fold serial dilutions were spotted on YPD plates without (CTRL) or with 0.8 M sodium chloride (NaCl). Growth was scored after 2–3 days. Growth on plates was quantified by using a quantitative imaging-based protocol described in Petropavlovskiy et al. [22]. Data were analyzed using unpaired Student’s *t*-test: * *p* < 0.01 was considered significant when comparing wild type versus hap4∆ cells at day 2 from three independent experiments.

**Figure 2 ijms-24-05320-f002:**
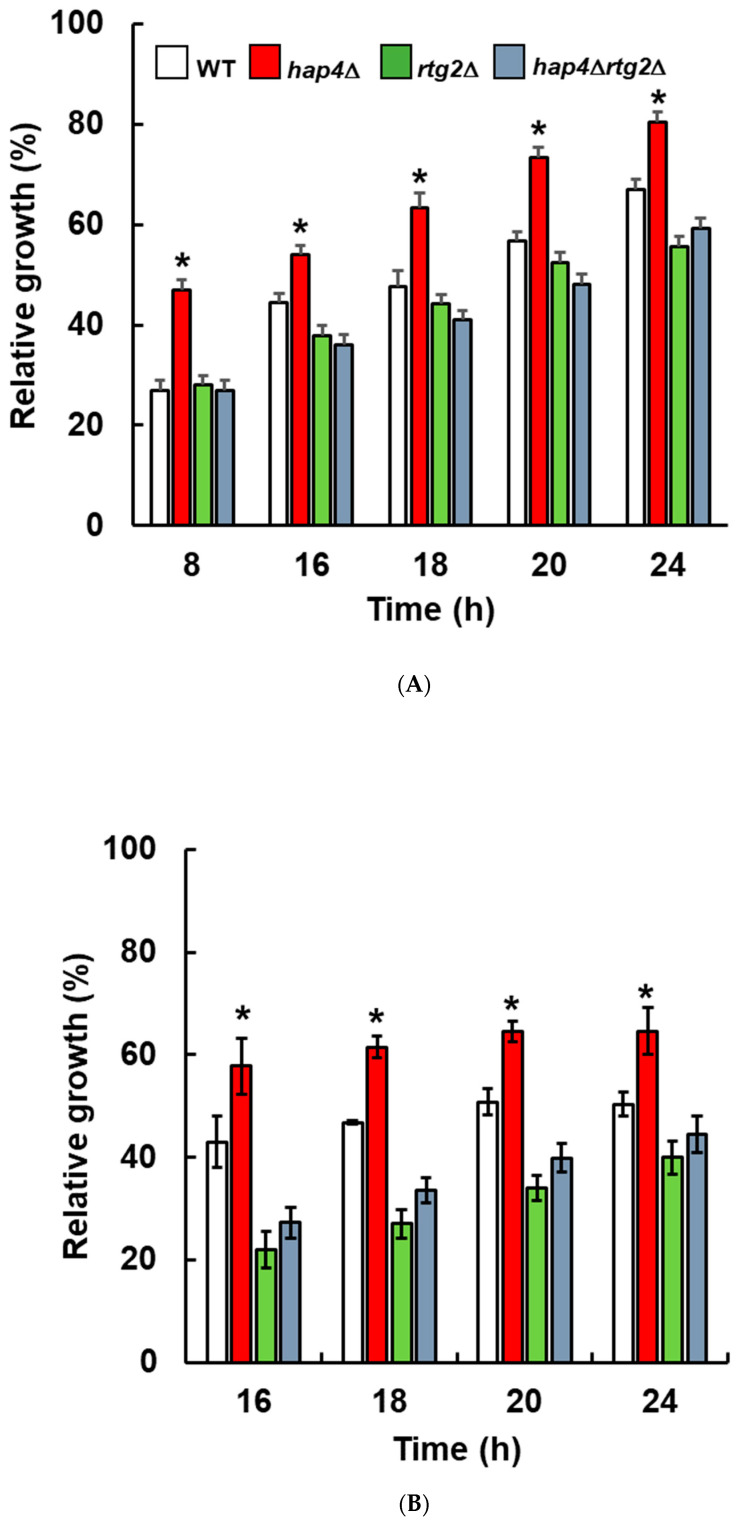
Relative growth of wild type and mutant cells in the presence of NaCl. (**A**) Wild type (WT) and indicated mutant cells (hap4∆, rtg2∆, hap4∆rtg2∆) grown overnight in YPD were diluted to 0.01 OD600 in fresh liquid YPD with or without 0.8 M sodium chloride (NaCl), and optical density was measured at 600 nm (OD600) over time with a high-precision TECAN microplate reader. (**B**) Wild type (WT) and indicated mutant cells (hap4∆, rtg2∆, hap4∆rtg2∆) grown overnight in YPD were diluted to 0.1 OD600 in fresh liquid YPD batch cultures with or without NaCl 0.8 M and optical density (OD600) was measured at the indicated times. In both (**A**,**B**), relative growth was calculated as the percentage of the OD600 of stressed/control cells. Unpaired Student’s *t*-test: a statistically significantly difference with * *p* < 0.01 when comparing wild type versus hap4∆ cells from three independent experiments.

**Figure 3 ijms-24-05320-f003:**
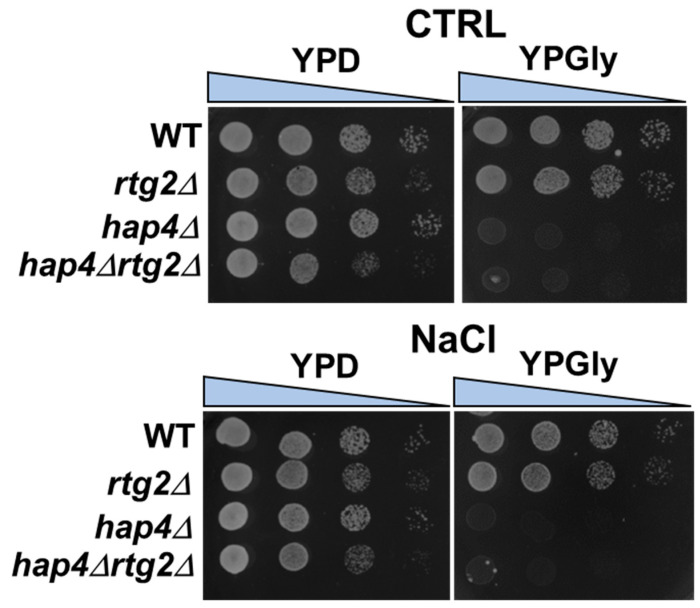
Mitochondrial respiratory competence upon NaCl stress. Wild type (WT) and mutant cells (rtg2∆, hap4∆, hap4∆rtg2∆) grown overnight in YPD medium were diluted to 0.1 OD_600_ in fresh liquid YPD with or without 0.8 M sodium chloride (NaCl). After 5 h, cells were diluted to 1 OD_600_ and ten-fold serial dilutions were spotted on YPD and YPGly plates. Growth was scored after 2–5 days. The images reported are representative of three independent experiments.

**Figure 4 ijms-24-05320-f004:**
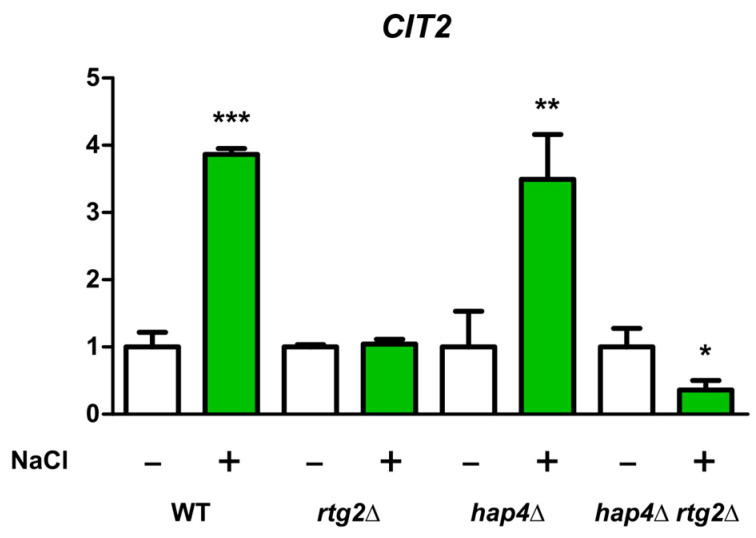
CIT2 expression under high osmotic environment. Wild type (WT) and mutant cells (hap4Δ, rtg2Δ, hap4Δrtg2Δ) grown overnight in YPD medium were diluted to 0.1 OD_600_ in fresh liquid YPD with or without 0.8 M sodium chloride (NaCl). After 5 h, cells were collected for RNA extraction, mRNA levels were measured by quantitative PCR, and the cells grown without NaCl stress were used as calibrator. The relative amount of CIT2 mRNA was calculated according to the comparative method (2^−ΔΔCt^). Values are mean ± SD of three independent experiments. Unpaired Student’s *t*-test: a statistically significantly difference with *** *p* < 0.0001; ** *p* < 0.01; * *p* < 0.05 when comparing untreated cells versus NaCl-stressed cells.

**Figure 5 ijms-24-05320-f005:**
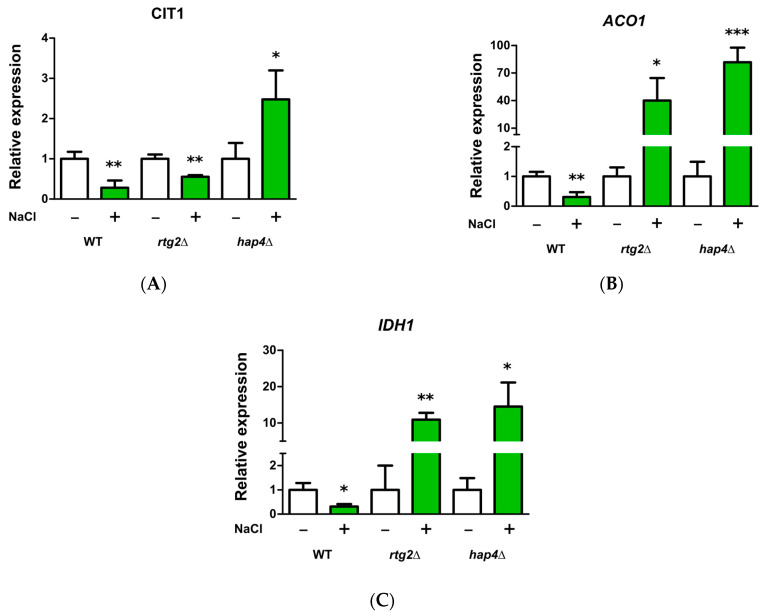
CIT1, ACO1 and IDH1 expression under high osmotic environment. Wild type (WT) and mutant cells (hap4Δ and rtg2Δ) grown overnight in YPD medium were diluted to 0.1 OD_600_ in fresh liquid YPD with or without 0.8 M sodium chloride (NaCl). After 5 h, cells were collected for RNA extraction, mRNA levels of CIT1 (**A**), ACO1 (**B**) and IDH1 (**C**) genes were measured by quantitative PCR, and the cells grown without NaCl stress were used as calibrator. The gene relative quantification was calculated according to the comparative method (2^−ΔΔCt^). Values are mean ± SD of three independent experiments. Unpaired Student’s *t*-test: a statistically significantly difference with * *p* < 0.05; ** *p* < 0.01; *** *p* < 0.001 when comparing untreated cells versus NaCl-stressed cells.

**Figure 6 ijms-24-05320-f006:**
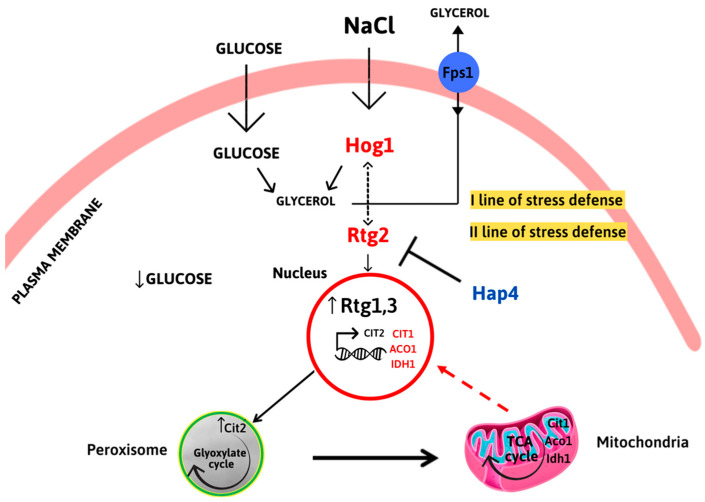
Interplay between *HAP4* and *RTG2* in osmoadaptation.

**Table 1 ijms-24-05320-t001:** Doubling time of wild type (WT) and mutant cells grown in the absence (control) and in the presence of sodium chloride (0.8 M NaCl). Reported data are the mean ± standard deviation of three independent experiments, each performed in triplicate.

Strains	Control Doubling Time (h)	+ NaCl Doubling Time (h)
**WT**	2.00 ± 0.32	3.90 ± 0.71
**hap4Δ**	2.55 ± 0.07	3.00 ± 0.42
**rtg2Δ**	2.20 ± 0.07	5.90 ± 0.12
**hap4Δrtg2Δ**	2.65 ± 0.07	5.95 ± 0.64

**Table 2 ijms-24-05320-t002:** Primers used for quantitative PCR.

Primer	Sequence
CIT1 Forward	5′-GCGCCTCCGAACAAACG-3′
CIT1 Reverse	5′-CTGCCTTTGCTGGGATAATTTC-3′
ACO1 Forward	5′-CAAGAACCCAGCTGACTATGACA-3′
ACO1 Reverse	5′-CCAATTCAGCTAGACCCAGAATATC-3′
IDH1 Forward	5′-CCCCTTCAATGTACGGTACCA-3′
IDH1 Reverse	5′-TGGACCACCGATCAAAGCA-3′
CIT2 Forward	5′-TGTAAGGCAATTCGTTAAAGAGCAT-3′
CIT2 Reverse	5′-CCCATACGCTCCCTGGAATAC-3′
ACT1 Forward	5′- ACTTTCAACGTTCCAGCCTTCT-3′
ACT1 Reverse	5′-ACACCATCACCGGAATCCAA-3′

## Data Availability

Raw data from this study are available upon reasonable request from the corresponding author.

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
