# Peer review of "Inactivation of HAP4 Accelerates RTG-Dependent Osmoadaptation in Saccharomyces cerevisiae"

_ijms, 2023, doi:10.3390/ijms24065320_

Round 1

Reviewer 1 Report

The roles of mitochondria in cellular signaling are well established, yet far from full understanding. Yeast Saccharomyces cerevisiae is exceptionally suited for mitochondrial studies, being able to grow despite the non-functional mitochondrial respiratory chain or in anaerobiosis. This adaptation is enabled, among others, by the RTG regulatory pathway enrolling peroxisomes to patch the incomplete TCA cycle in respiratory deficient yeast cells. Previously, the authors demonstrated HOG pathway-dependent participation of RTG signaling in response to osmotic stress. In the reviewed manuscript, the authors studied the involvement of the HAP2-5 complex, another master regulator of mitochondrial functions, and RTG signaling in osmotic stress response. They found that the rtg2Δ strain is oversensitive to salt stress, whereas the hap4Δ strain is transiently slightly more resistant to salt stress than the wild-type (WT) strain. They demonstrated that the absence of the HAP4 gene does not affect salt stress induction of the RTG-controlled genes measured as CIT2 up-regulation. Finally, they showed that salt stress in the WT strain causes strong repression of CIT1, ACO1, and IDH1 genes encoding three TCA enzymes. Conversely, in rtg2Δ or hap4Δ strains, ACO1 and IDH1 genes are strongly induced, whereas CIT1 is moderately induced only in the hap4Δ strain. The authors conclude that HAP4 and RTG2 “reciprocally interact to control mitochondrial function upon salt stress.”

Despite the significant issues listed below, the results described in this manuscript foreshadow some potentially interesting phenomena. Unfortunately, the study is too preliminary, and further investigation is needed to gather additional data that would help to get a deeper insight into these phenomena and make the story more complete.

Major points:

1.

There is a considerable disagreement between the drop test results shown in Figures 1 and 3. The drop test description in both figure legends is the same, and the test was done on the same strains, but the images for YPD-grown cells with and without 0.8 M NaCl are quite different. First, the upper and lower images of the YPD plate in Figure 3 are identical, which is a major fault. Do they both depict YPD control or YPD+NaCl plate? In any case, they do not resemble the images of the YPD control or YPD+NaCl plate in Figure 1. If the image in Figure 3 depicts the YPD+NaCl plate, then, contrary to Figure 1, the oversensitivity of rtg2Δ strain to salt stress is barely visible, and we do not see higher than WT resistance of hap4Δ strain to this stress. If the image in Figure 3 depicts the YPD control plate, then it looks like it was taken after one day of growth, and we see slightly slower growth of rtg2Δ strain, which we do not see on the respective image in Figure 1.

2.

The title of the subchapter: “2.1.3. RTG pathway is up-regulated in cells lacking HAP4”, is misleading. It suggests that the absence of the HAP4 gene causes induction of the RTG pathway, whereas in fact (Figure 4), the data for hap4Δ and WT strains are virtually identical. So the absence of the HAP4 gene does not affect salt stress induction of the RTG pathway.

3.

The legend to Figure 5 (but not the relevant part of the manuscript) announces the data for rtg2Δ hap4Δ double deletion mutant, which are not shown in this Figure.

4.

Figures 4 and 5: The authors show the changes in CIT2, CIT1, ACO1, and IDH1 expression in WT and rtg2Δ or hap4Δ cells under salt stress relative to unstressed cells of the same strains. It would be interesting to see how the absence of RTG2 or HAP4 genes influences the expression of CIT2, CIT1, ACO1, and IDH1 in unstressed cells. Indeed, in the authors’ previous paper (cited as ref. 12), the expression of the CIT2 gene in WT and rtg2Δ strains grown in YPD with or without 0.8 M NaCl was depicted differently than in the reviewed manuscript. Intriguingly, the message carried by that previous result is also different. In the reviewed manuscript, the result shown in Figure 4 indicates that non-functional RTG signaling prevents the CIT2 up-regulation by the salt stress but has no influence on the CIT2 basal expression. Conversely, Figure 5 in the previous paper indicates that in the YPD-grown rtg2Δ strain, the expression of CIT2 is strongly reduced compared to the YPD-grown WT strain. However, in both strains, CIT2 is up-regulated by salt stress to approximately the same extent. Consequently, if, as the authors assume, CIT2 expression is the measure of RTG signaling, then RTG signaling is not involved in response to salt stress. The authors should clarify which one of these alternatives is true.

5.

The usage of the word “faster” (lines 235 and 246) is ungrounded. The authors did not demonstrate the kinetics of the salt stress response. They only showed that:

- rtg2Δ strain is oversensitive to salt stress compared to WT,

- CIT2 induction by salt stress depends on RTG2 (but see previous remark),

- In the rtg2Δ strain, there is a strong induction of ACO1 and IDH1 by salt stress, whereas salt stress represses these genes in the WT strain.

6.

The results presented in Figure 5 imply that when both RTG and HAP regulatory systems are functional, salt stress induction of ACO1 and IDH1 is prevented. This observation is intriguing and requires further investigation to discover what phenomenon lies behind this observation.

7.

Lines 254-256 – The authors refer to their not shown data that hap4Δ strain can grow in the absence of glutamate, whereas rtg2Δ can not. rtg2Δ glutamate auxotrophy has been known for many years. The ability of hap4Δ or WT strain to grow in the absence of glutamate is not sufficient to support the hypothesis proposed in the preceding sentence.

8.

The model shown in Figure 6 carries no meaningful message and is not adequately supported by the results presented in the manuscript. In particular, the statement: “HAP4 acts as a negative regulator of RTG-dependent expression of TCA cycle genes CIT1, ACO1, IDH1.” (lines 283-284). The authors demonstrated that in both rtg2Δ and hap4Δ strains, ACO1 and IDH1 genes are up-regulated. No data regarding HAP4 and RTG2 interplay are presented. The model is a slight modification of the model already published in a previous paper by the same authors (ref. 12).

Minor remarks:

9.

Please adjust gene and allele names according to conventions established for S. cerevisiae. Gene names should be written uppercase and italicized. Mutated or null alleles should be written lowercase and italicized. Delta character should be at the end of the null allele name, i.e., hap4Δ.

10.

Saccharomyces cerevisiae and S. cerevisiae should be italicized.

11.

The authors alternatively use “oxoglutarate” and “ketoglutarate”. Current nomenclature rules recommend “oxoglutarate”. Please unify.

12.

Spelling and editing errors:

The authors alternatively use “signaling” or “signalling”. Please unify according to the spelling variant selected for the whole manuscript, which appears to be American English.

Line 160 – replace “supression” with “suppression”

Line 180 – the phrase “up-regulated in NaCl” should be rephrased to make it more precise.

Lines 197-201 – This sentence should be rephrased for clarity.

Line 200 – replace “osmodaptation” with “osmoadaptation”

Line 245 – “stress specific and strictly RTG-dependent up-regulation could be observed for CIT1” Did the authors mean CIT2?

Line 288 – replace S. cerevisiae with Saccharomyces cerevisiae because this is the first instance of budding yeast systematic name in the manuscript.

Subchapter “4.4. Quantitative PCR (qPCR)” needs thorough editing.

Author Response

Please, Kindly find attached our response.

Author Response

Please, kindly find attached our response.

Reviewer 3 Report

The manuscript entitled “Inactivation of HAP4 accelerates RTG-dependent 2 osmoadaptation in yeast” is significant in this field of interest. The manuscript is well-structured with enough data. This manuscript has a few issues needed to be addressed. Thus I recommend this manuscript for minor revision.

Kindly check the typographical errors in the entire manuscript.

Particularly in the methods and material section “The S. cerevisiae strains used in this study were W303-1B (WT) cells (MATα ade2 leu2 his3 trp1 ura3) and derivatives hap4 (hap4::KanMX4), rtg2 (rtg2::LEU2), hap4rtg2 (rtg2::LEU2 hap4::KanMX4) [9].  Cells were grown at 30 ° C in YPD (1% yeast extract, 2% bactopeptone and 2% glucose with 2% agar for solid medium) in the absence or in the 291 presence of sodium chloride (NaCl)." and section “4.4 Quantitative PCR (qPCR)”

Kindly justify the use of 0.8M NaCl for this study.

In figures 2 C and B, Y axis scales should be at the same level. Kindly make it to 100%.

While comparing the micro and macro culture, the relative growth (%) of the HAP4 (cells) appeared to be static in all-time duration in the macro culture method. Even though, in macro culture HAP4 was increased seems to be a not significant difference in the all-time point of analysis. Whereas the growth of HAP4 was increased in the micro-dilution significantly. kindly justify the reasons.

Author Response

(The authors gave the same response as above.)

Reviewer 4 Report

The authors in the present study analyzed the interrelationship between RTG2 and HAP4 in a Saccharomyces cerevisiae model. They showed that the inactivation of HAP4 improved the kinetics of osmoadaptation by eliciting both the activation of retrograde signaling and the upregulation of three TCA cycle genes, CIT1, ACO1, and IDH1. 

The following recommendations are then made to the authors to improve their manuscript.

Abstract

1.    Authors are encouraged to define the abbreviations used.

Introduction

1.    Authors are encouraged to define the abbreviations used.

2.    The authors are encouraged to provide more information on how HAP4 regulates mitochondrial function and how it may regulate gene expression between the nucleus and mitochondria.

3.    In section 4.1, it was mentioned which dilutions of NaCl were used.

4.    Authors are encouraged to add a section on mitochondrial respiratory assessment to explain how this parameter was assessed.

5.    In section 4.2, the phrase "was calculated" is duplicated in line 305.

Materials and Methods

1.    In section 4.1, write S. cerevisiae in italics or underlined.

2.    Section 4.1 mentions the used concentration of NaCl.

3.    In section 4.3, mention which algorithm is referred to in line 323.

4.    It is recommended that the authors in section 4.4 adapt the primers used in a table and mention which are mitochondrial and nuclear.

Results

1.    In Figures 1 and 3 indicate what each of the points refers to.

2.    Authors are encouraged to standardize the style of Figure 2.

3.    In Figure 2B, clarify which values are referred to in the columns "control" and "NaCl."

4.    Authors are requested to include graphs of mitochondrial respiration in section 2.1.2 and the analysis of these respiratory data.

5.    Indicate in figure 3 the number of replicates

Discussion

1.    In lines 230 and 231, the authors could shed more light on how HAP4 coordinates the interaction between the nucleus and the mitochondrion.

Author Response

(The authors gave the same response as above.)

Reviewer 5 Report

This is a review for 'Inactivation of HAP4 accelerates RTG-dependent osmoadaptation in yeast'

The manuscript is well-written and the results are clearly presented. However, there are a few minor issues with the title and methods. The methods should have more information to enable others to reproduce the work.

Title: Huge supposition. The work was carried out only on S. cerevisiae and not on different types of yeasts. Not even different species of Saccharomyces. Remove yeasts from the title and replace it with Saccharomyces cerevisiae

Introduction: There is little specific information on recent work/reviews on HAP4, osmoadaptation, and RTG2. Up to 6 new references (2 on each) should be added.

Fig 1: Why is the bar chart on density not shown?

Will a higher initial density affect stress response?

Line 288: S. cerevisiae should be in italics

Line 288: Where was W303-1B (WT) strain obtained? Source? The constructs/mutants as well?

Some media sources (Manufacturers, country were not stated e.g YPD

Line 300: 'fresh overnight pre-cultures'. Overnight is ambiguous. Provide the estimated time in hours cell growth lasted.

Line 313:' serial dilutions were spotted' What amount of culture was spotted?

Line 365: Put Microsoft before Excel

Author Response

(The authors gave the same response as above.)

Round 2
